# Bond Behavior of Steel Rebar Embedded in Cementitious Composites Containing Polyvinyl Alcohol (PVA) Fibers and Carbon Nanotubes (CNTs)

**DOI:** 10.3390/polym15040884

**Published:** 2023-02-10

**Authors:** Dongmin Lee, Seong-Cheol Lee, Sung-Won Yoo

**Affiliations:** 1Department of Civil Engineering, Kyungpook National University, 80, Daehak-ro, Buk-gu, Daegu 41566, Republic of Korea; 2Department of Civil and Environmental Engineering, Gachon University, Seongnam 13120, Republic of Korea

**Keywords:** bond behavior, cementitious composites, carbon nanotubes (CNTs), bond stress-slip model, polyvinyl alcohol (PVA)

## Abstract

In this study, pull-out tests were conducted to investigate the bond behavior of a rebar embedded in cementitious composites with polyvinyl alcohol (PVA) fibers and carbon nanotubes (CNTs). In the cementitious composites, the binder consisted of ordinary Portland cement, blast furnace slag, and fly ash, with a weight ratio of 39.5, 21.0 and 39.5%, respectively, while the nonbinder consisted of quartzite sand, lightweight aggregate, superplasticizer, and shrinkage-reducing admixture. The water/binder ratio and volume fractions of the PVA fibers were 32.9% and 2.07%, respectively. In the test program, the rebar diameter (D13, D16, and D19) and CNTs mix ratio (0.0, 0.1, 0.2, and 0.3 wt.%) were considered as the test variables. The test results showed that the bond strength of a rebar increased as the rebar diameter decreased or as the CNTs mix ratio increased. Based on the test results, a new, simple model has been proposed with consideration of the rebar diameter, as well as the CNTs mix ratio. Comparing the test results, it was investigated that the proposed model generally represented the bond behavior well, including the bond strength and the corresponding slip of a rebar embedded in PVA cementitious composites, with or without CNTs.

## 1. Introduction

Due to the demand for securing the safety of large structures, many studies are being conducted to improve the structural performance of reinforced concrete. In line with these demands, research on fiber-reinforced concrete has been actively conducted to overcome the shortcomings of concrete exhibiting brittle behavior after cracking. Owing to fibers bridging a crack, fiber-reinforced concrete can exhibit ductile behavior, even after cracking, with either tension softening behavior or strain hardening behavior [1]. In addition to improving the concrete performance after cracking, research on improving the concrete matrix itself has recently been conducted by adding carbon nanotubes (CNTs) to the concrete mixture. Recently, several types of research have been conducted to utilize the advantages of fibers and CNTs together. However, most of the research has focused on the material properties of fiber-reinforced concrete with CNTs.

To use fiber-reinforced concrete with CNTs as a structural member, it is necessary to investigate the interaction between the reinforcing bars and fiber-reinforced concrete with CNTs. In this study, as a part of the research for the structural behavior of CNTs-mixed fiber-reinforced concrete with reinforcing bars, the bond behavior of the steel rebar embedded in the PVA cementitious composites with CNTs is investigated through an extensive pull-out test program. Based on the test results, a conventional model for the bond stress-slip behavior of a rebar is modified to reflect the effect of PVA cementitious composites with CNTs. The test results and the proposed model will help the investigations of the effect of CNTs on structural behaviors, such as rebar development length, tension-stiffening behavior, etc.

## 2. Literature Review

### 2.1. Literature Review on Fiber-Reinforced Concrete

To apply fiber-reinforced concrete as a structural member, several models have been developed to describe the material behaviors. Some research groups have developed a tensile stress-crack width relationship to represent the tensile behavior of fiber-reinforced concrete. Martie et al. [2] proposed the tensile model for fiber reinforced concrete, based on the uniform bond stress along a fiber. Voo and Foster [3] proposed the Variable Engagement Model (VEM) by introducing the engagement length of a fiber so that the tensile behavior of concrete reinforced with straight steel fibers was reasonably predicted. Leutbecher and Fehling [4] presented a model considering the effect of fibers on crack widths in steel fiber-reinforced concrete containing rebars. Stroeven [5] developed a formulation that could consider fiber type on the tensile behavior. Lee et al. [6,7] proposed the Diverse Embedment Model (DEM), considering the mechanical anchorage effect of end-hooked steel fibers, as well as the frictional bond behavior of fibers. Later, the DEM was simplified by eliminating the double numerical integration in the DEM [8]. Meanwhile, others have proposed compressive stress-strain relationships to represent the compressive behavior of fiber-reinforced concrete. Ezeldin and Balaguru [9] presented a compression model for fiber reinforced concrete based on the test results. Hsu and Hsu [10] also proposed an empirical equation to describe the compressive behavior of fiber-reinforced concrete. Someh and Saeki [11] proposed a model for steel fiber-reinforced concrete under compression. Mansur et al. [12] conducted the compression test for high-strength fiber-reinforced concrete. Nataraja et al. [13] derived a simple analytical model to describe the compressive stress-strain response. Experimental programs [14,15,16] have also been actively conducted to investigate the tensile behavior of fiber-reinforced concrete with reinforcing bars. Based on the test results, Lee et al. [17] proposed the tension stiffening model, which was beneficial for analyzing the tensile behavior of fiber-reinforced concrete members with reinforcing bars. Recently, with the development of a rigorous analysis procedure, it has become possible to predict the nonlinear structural behavior of fiber-reinforced concrete members or structures [18,19,20].

### 2.2. Literature Review on Cementitious Composites with CNTs

Types of CNTs are generally divided into two groups: single-walled CNTs (SWCNTs) and multi-walled CNTs (MWCNTs). Kang et al. [21] showed the differences between SWCNTs and MWCNTs in their characteristics, including diameter, length, modulus of elasticity, tensile strength, electrical conductivity, and heat conductivity, as shown in Table 1. The table shows that MWCNTs were relatively better in terms of their tensile strength than SWCNTs, while SWCNTs were better in thermal and electrical conductivity than MWCNTs. Many studies [22,23,24,25,26] also investigated the effect of CNTs on cement composite or concrete performance when CNTs were additionally mixed. Silvestro and Gleize [22] reviewed the existing studies regarding the effect of CNTs on the compressive strength and flexural strength of cement-based material. They investigated, through the literature, whether incorporating CNTs increased the strength of cement-based materials if the CNTs were adequately dispersed. Through an experimental program, Cerro-Prada et al. [23] showed that the flexural and compressive strength increased as the MWCNTs mix ratio increased. Amin et al. [24] investigated, through an experimental program, that the addition of 0.1 wt.% of CNTs showed an improvement in the thermal and mechanical properties of the hardened Homra/OPC blended cement composites. Zhang et al. [25] investigated the effect of CNTs on the thermal and electrical conductivity of cementitious composites. They showed that the thermal resistivity increased up to a specific temperature as the CNTs were additionally mixed. Nam et al. [26] showed that the EM-wave-shielding performance was enhanced when MWCNTs were mixed into the cement matrix.

Lee et al. [27] stated that the mechanical properties of concrete with steel fibers and CNTs were superior to those with only steel fibers or CNTs. Azhari and Banthia [28] noted that the electrical conductivity of carbon fiber-reinforced cementitious composites with MWCNTs considerably increased, and similar results have been reported by Park et al. [29]. Jang et al. [30] investigated the strain-detecting of cementitious composite with synthetic polyethylene (PE) and steel fibers. They found that adding MWCNTs improved the self-sensing performance for the strain measurements. Lee et al. [31] showed that the compressive behavior of polyvinyl alcohol (PVA) fiber-reinforced concrete was enhanced by adding CNTs to the mixture. Nuaklong et al. [32] demonstrated significant improvements in the residual compressive and flexural strengths around the heating temperature of 400 ℃ for cementitious composites with both PP fibers and MWCNTs.

### 2.3. Literature Review on the Bond Behavior of a Rebar

Many studies [33,34,35,36,37] have proposed simple models to represent the bond stress-slip behavior of deformed steel rebars embedded in ordinary concrete. This section reviews and discusses the conventional bond stress-slip models, followed by their applicability to PVA cementitious composites with CNTs. It is noted that bond behavior with the pull-out failure is discussed here because no splitting crack was observed in the test program of this study.

#### 2.3.1. CEB-FIP Model Code 2010 [37]

The CEB-FIP Model Code 2010 (MC10) [37] adopted the bond stress-slip model for the deformed steel rebar proposed by Eligehausen et al. [35]. The MC10 bond stress-slip model consists of three phases: an ascending branch, plateau, and descending branch. For the case with a good bond condition under a pullout failure, each phase is expressed as the following equations:(1a)$$\tau =\tau_{max}{\left(\frac{s}{s_{1}}\right)}^{\alpha }\;\mathrm{for}\;0\leq s\leq s_{1}$$
(1b)$$\tau =\tau_{max}\;\mathrm{for}\;s_{1}\leq s\leq s_{2}$$
(1c)$$\tau =\tau_{max}-\left(\tau_{max}-\tau_{f}\right)\frac{\left(s-s_{2}\right)}{\left(s_{3}-s_{2}\right)}\;\mathrm{for}\;s_{2}\leq s\leq s_{3}$$
(1d)$$\tau =\tau_{f}\;\mathrm{for}\;s_{3}\leq s$$
where $\tau_{max}=2.5\sqrt{f_{cm}}$ MPa, $\tau_{f}=0.4\tau_{max}$, $s_{1}=1.0\;\mathrm{mm}$, $s_{2}=2.0\;\mathrm{mm}$, $s_{3}$ is the clear distance between ribs, α=0.4, and $f_{cm}$ is the mean of the concrete compressive strength.

#### 2.3.2. Soroushian et al. [34]

Soroushian and Choi [33] conducted the pullout tests for a rebar embedded in confined concrete. The test program considered 16 mm, 22 mm, and 25 mm diameter deformed steel rebars embedded in concrete with a compressive strength of 30 MPa. Through the tests, they investigated whether the bond strength decreased as the rebar diameter increased. With the additional pullout test to consider high-strength concrete, Soroushian et al. [34] proposed the bond stress-slip model for the ascending part with the following equation:(2)τ=τmax(ss1)·e[1−(ss1)α]/α for 0≤s≤s1
where $\tau_{max}=20-\frac{d_{b}}{4}\sqrt{\frac{f_{c}^{\prime }}{30}}$ MPa, $\tau_{f}=5\;\mathrm{MPa}$ $s_{1}=1.0\;\mathrm{mm}$, $s_{2}=3.0\;\mathrm{mm}$, $s_{3}=10.5\;\mathrm{mm}$, and *α* = 0.4.

In contrast to the MC10 model, this model considered the rebar diameter in the bond strength.

#### 2.3.3. Harajili et al. [36]

Harajili et al. [36] proposed the bond stress-slip model for a rebar embedded in plain or fiber-reinforced concrete. In contrast to the models above, they proposed the model considering the effect of the rebar diameter on the slip. They adopted the same format as the MC10 model, but they considered different values, with $\tau_{max}=2.57\sqrt{f_{cm}}$ MPa, $\tau_{f}=0.9\sqrt{f_{cm}}$ MPa, $s_{1}=0.15c_{O}$ mm, $s_{2}=0.35c_{O}$ mm, $s_{3}=c_{o}$ mm, α=0.3, and $c_{o}$ is the clear distance between the ribs.

#### 2.3.4. Comparison of Existing Models

Figure 1 compares the existing models for the bond strength and its corresponding slip for a rebar embedded in concrete. To investigate the effect of the concrete compressive strength, 24 and 40 MPa of concrete compressive strengths were considered in the comparison. As the figure shows, Soroushian et al. presented higher bond strength than the other two models when the rebar diameter was smaller than 20 mm. They reported that the bond strength decreased as the rebar diameter increased. On the other hand, MC10 and Harajili et al. did not reflect the effect of the rebar diameter on the bond strength. For the slip corresponding to the bond strength ($s_{1}$), the effect of the concrete compressive strength was ignored in the tree models. Regarding the effect of the rebar diameter, Harajili et al. considered that the $s_{1}$ increased as the rebar diameter increased, while the other two models presented that the $s_{1}$ was constant to 1.0 mm regardless of the rebar diameter.

The comparison of the bond stress-slip responses until the bond strength is reached is depicted in Figure 2. To standardize the results, the bond stress and slip are normalized by the bond strength and its corresponding slip, respectively. The results show that the MC10 model and the model proposed by Harajili et al. exhibit similar trends in their curve shapes. However, the results of Harajili et al. reveal higher bond stress, with α = 0.3, which is a lower value than the α value of 0.4 in the MC10 model. On the other hand, the model proposed by Soroushian et al. displays a less steep initial slope compared to the other models, but exhibits higher bond stress values beyond the range of *s*/*s*_1_ = 0.1~0.2.

## 3. Research Significance

Recently, several types of research have been conducted to investigate the material behavior of cementitious composites with fibers and CNTs together in order to take all of the advantages before and after cracking. However, there are few studies on the bond behavior of a rebar in cementitious composites containing both fibers and CNTs. In this study, an experimental program was conducted to investigate the bond behavior of a rebar embedded in PVA cementitious composites with CNTs. From the pull-out test, the bond stress-slip behavior was measured so that the effect of the rebar diameter and CNTs mix ratio could be evaluated. Based on the test results, a new simple bond stress-slip model has been proposed. As the interaction between the cementitious composites and the rebar is significant in the structural behavior, the test results and the proposed model will help to predict the structural behavior of PVA cementitious composites with CNTs.

## 4. Test Program

### 4.1. Materials

The mix proportions of the PVA cementitious composites used in this study are summarized in Table 2. In the mix proportions, the binder consisted of Ordinary Portland Cement (OPC), blast furnace slag (BFS), and fly ash (FA) with a weight ratio of 39.5%, 21.0%, and 39.5%, respectively. The nonbinder consisted of quartzite sand (particle size 0.1~1.7 mm), lightweight aggregate (LA), superplasticizer (SP), and shrinkage-reducing admixture (SRA). The water/binder ratio and volume fractions of the PVA fibers were 32.9%, and 2.07%, respectively, which were constant throughout all of the test specimens. The geometric configuration and mechanical properties of the PVA fibers are presented in Table 3. To investigate the effect of the CNTs on the bond behavior of a rebar, four CNTs mix ratios were considered, from 0.0 wt.% to 0.3 wt.% to the binders, referred to in the previous studies [38,39,40,41,42]. The mix proportions were named CNT-0.0 through CNT-0.3 according to the CNTs’ mix ratio.

In this study, MWCNTs were used as they are easier to produce, relatively inexpensive, and more dispersed than SWCNTs [38,43]. In addition, MWCNTs have better mechanical properties than SWCNTs [44]. To suppress aggregation by van der Waal`s force and achieve the uniform dispersion of the CNTs, a 5% polycarboxylate superplasticizer was added to the 3% CNTs aqueous solution and sonicated, as presented in the literature [38,45].

Three rebar sizes of D13, D16, and D19 were considered in the test program to investigate the effect of a rebar size. All of the rebars have a grade of SD400 according to the Korean standard (KS) [46]. The geometric properties of the rebar, including the cross-sectional area, perimeter, and the distance and height of the ribs, are presented in Table 4.

### 4.2. Pullout Test Specimens

To investigate the bond behavior of a deformed steel rebar embedded in the PVA cementitious composites with CNTs, the pullout test was conducted considering the test variables, the rebar diameter and the CNTs mix ratio. For the pull-out test, 150 × 150 × 200 mm prismatic specimens were fabricated, as shown in Figure 3. As presented in the figure, the embedment length was two times the rebar diameter placed in the specimen’s center. The PVC pipe was implemented to make the unbonded part of the rebar, as the previous studies adopted for the pull-out test [47,48,49].

### 4.3. Fabrication

When the specimens were fabricated, the cementitious composites were first dry-mixed. Then, water, 3% CNTs aqueous solution, and PVA fibers were added and mixed in the order presented in Figure 4. It is noted that the mix order was referred to in the previous study [31], so that the workability was attained even after the CNTs were added.

The pullout test specimens were fabricated following the order presented by Yoo and Shin [50]. First, the molds were made with wood, and holes were drilled in the middle of both sides in order to place the steel rebar. When the steel rebar was placed in the mold, the embedment length was set to two times the rebar diameter by covering the unbonded parts with PVC pipes. The embedded part was placed in the center of the specimen. Then, the PVA cementitious composites with CNTs were filled in the mold. Three specimens were fabricated for each test variable to obtain reliable test results. ϕ100 × 200 mm cylindrical specimens, to measure the compressive behavior, were also prepared in the same order. All of the specimens were vibrated on a vibration table, then dry-cured for 28 days.

Table 5 presents the slump and slump flow measured after mixing the PVA cementitious composites with CNTs. As presented in the table, both the slump and slump flow decreased as the CNTs mix ratio increased. It can be inferred that the lower workability was mainly due to the high surface area and surface tension property of the CNTs [51,52,53].

### 4.4. Pull-Out Test Set-Up and Procedure

Figure 5 shows the setup for the pull-out test. As presented in the figure, two linear variable differential transducers (LVDTs) were attached to the free end of the rebar and the top surface of the specimen to measure the free-end slip. In contrast, one LVDT was attached to the rebar at the loaded end. The pull-out load was applied to the rebar at a rate of 0.5 mm/min using a 2000 kN capacity universal testing machine (UTM) to ensure static loading. The pullout test was conducted until the applied load decreased to 50% of the peak load due to the limitation of the test equipment configuration.

## 5. Test Results

### 5.1. Material Properties

Table 6 presents the material properties of the PVA cementitious composites with CNTs. It is noted that the material properties were measured through the compression test, which followed the procedure presented in ASTM C39 [54]. During the compression tests, the strains were measured through two LVDTs attached to the side of the cylindrical specimen so that the modulus of elasticity could be evaluated from the stress-strain responses, as presented in Figure 6a. As presented in the table, it was observed that the compressive strength and the modulus of elasticity generally increased as the CNTs ratio increased, with the exception of CNT-0.1. On the other hand, the strain corresponding to the compressive strength was not much affected by the CNTs mix ratio. The direct tension test results of the rebars are presented in Figure 6b. The yield strength and the elastic modulus are evaluated from the stress-strain responses, as presented in Table 6.

### 5.2. Pullout Test Results

#### 5.2.1. Failure Mode

Figure 7 shows one of the specimens after the pull-out test. As seen in the figure, all of the specimens exhibited pull-out failure, with neither a splitting crack nor rebar yielding. This is mainly due to the short embedment length of the rebar, the sufficiently thick cementitious composite cover thickness, and the PVA fibers.

#### 5.2.2. Bond Stress-Slip Responses

From the load-slip responses, measured through the pull-out test, the bond stress-slip responses were evaluated. The following equation evaluated the bond stress:(3)$$\tau =\frac{P}{\pi d_{b}l_{b}}$$
where τ is bond stress (MPa), P is applied load (N), $d_{b}$ is the rebar diameter (mm), and $l_{b}$ is the embedment length (mm). It is noted that the slip measured at the loaded end was considered in the same manner as conducted by Wardeh et al. [55] because the slip at the free end exhibited inconsistent results.

Figure 8 shows the bond stress-slip responses of a rebar evaluated from the pull-out test results. When one of the three test results was too scattered from the other two, it was excluded from the analysis of the test results. As seen in the figure, the bond stress-slip responses could appear to be divided into three phases: ascending, plateau, and descending. However, some specimens exhibited a plateau of very short duration. 

#### 5.2.3. Bond Strength

The bond strengths ($\tau_{\max}$) evaluated from the bond stress-slip responses are presented in Table 7 and Figure 9. It is noted that the bond strengths were evaluated to the average of the test results of two or three specimens. As seen in the table and figure, the bond strength increased as the CNT mix ratio increased, with the exception of CNT-0.10, which exhibited relatively low compressive strength. Therefore, it can be seen that the bond strength is highly affected by the compressive strength of PVA cementitious composites. In addition, the bond strength decreased as the rebar diameter increased. This tendency was consistent with the test results presented by many studies [52,56] and the analytical model [33].

#### 5.2.4. Slip Corresponding to the Bond Strength

The slips corresponding to the bond strengths ($s_{1}$) are presented in Table 7 and Figure 10. As compared in the table and figure, except for the specimens without CNTs, the slip corresponding to the bond strength generally increased as the rebar diameter increased, which was consistent with the test results presented by many other studies [52,56] and the analytical model [33]. On the other hand, it was observed that the effect of the CNTs mix ratio on the slip upon the bond strength was not consistent.

#### 5.2.5. α Coefficient

To represent the ascending part of the bond stress-slip response, several models [35,36,37] adopted the type of the following equation:(4)$$\tau =\tau_{max}{\left(\frac{s}{s_{1}}\right)}^{\alpha }\;\mathrm{for}\;0\leq s\leq s_{1}$$
where α is a coefficient for the shape of the ascending curve.

Table 7 and Figure 11 present the coefficient α, which was evaluated through regression with the least square error when comparing the test results and equation (4) for the ascending part. It is noted that it is chosen between 0 to 1; a smaller value is closer to higher initial stiffness in the bond stress-slip response. As the table and figure shows, the effect of the CNTs mix ratio and the rebar diameter could have been more obvious on the α coefficient. The average value for the entire test results was calculated at 0.42, slightly larger than the 0.4 generally adopted for the bond behavior of a deformed rebar embedded in ordinary concrete [37]. If the test result with the D19 rebar and no CNTs is excluded, the coefficient was averaged to 0.38. Therefore, the coefficient for the PVA cementitious composite with CNTs can be similar to that for ordinary reinforced concrete.

## 6. Proposed Model and Its Verification

### 6.1. Proposed Model

As the test results showed that the bond strength of a rebar in PVA cementitious composite with CNTs decreased with the increase in the rebar diameter, this study assumes that the bond strength is inversely proportional to the rebar diameter. With the consideration of the square root of the concrete compressive strength, similar to the MC10 [37] and Harajili’s models [36], the following equation has been developed:(5)$$\tau_{max}=\frac{28}{d_{b}}\sqrt{f_{cm}}\;\mathrm{MPa}$$

It is noted that the coefficient 28 in the above equation was derived through regression based on the least square error method for the differences in the bond strengths between the model predictions and the test results.

This study adopted four piecewise equations similar to MC10 and Harajili’s models to represent the bond stress-slip relationship. The proposed bond stress-slip model is expressed with the following equations:(6a)$$\tau =\tau_{max}{\left(\frac{s}{s_{1}}\right)}^{\alpha }\;\mathrm{for}\;0\leq s\leq s_{1}$$
(6b)$$\tau =\tau_{max}\;\mathrm{for}\;s_{1}\leq s\leq s_{2}$$
(6c)$$\tau =\tau_{max}-\left(\tau_{max}-\tau_{f}\right)\frac{\left(s-s_{2}\right)}{\left(s_{3}-s_{2}\right)}\;\mathrm{for}\;s_{2}\leq s\leq s_{3}$$
(6d)$$\tau =\tau_{f}\;\mathrm{for}\;s_{3}\leq s$$

In the above equation, based on the test results in this study, it was suggested that $\tau_{f}=0.4\tau_{max}$ MPa, $s_{1}=0.08c_{O}$ mm, $s_{2}=0.12c_{O}$ mm, $s_{3}=c_{O}$ mm, and α=0.4. As a result of the test in this study, $s_{1}$ and $s_{2}$ are smaller than those of Harajili et al. [36], considering that the slip corresponding to the bond strength and the plateau was smaller for the rebar embedded in PVA cementitious composites with CNTs than those of the ordinary reinforced concrete.

### 6.2. Verifications

The bond strengths predicted by the proposed model have been compared with the test results, as presented in Table 8 and Figure 12. The previous models have also been compared in the table and figure. As presented in the table and figure, it was investigated that the previous models significantly overestimated the bond strength of the rebar embedded in the PVA cementitious composites with or without CNTs. The ratio of the predicted values to the test results averaged as 1.42, 1.46, and 1.66 for MC10, Harajili et al., and Soroushian et al., respectively. It is inferred that the previous models overestimated the bond strength because there was no coarse aggregate in the PVA cementitious composites. On the other hand, the proposed model showed good agreement with the test results, with a mean of 1.02 and a standard deviation of 0.24 for the ratio of the predicted values to the test results. Although there was some deviation from the test results, the proposed model showed a much smaller standard deviation than the previous models. In more detail, the proposed model reflected the effect of the rebar diameter on the rebar bond strength well. In the proposed model, the effect of the CNTs’ mix ratio on the bond strength was taken into account with the concrete compressive strength. Therefore, it can be concluded that the proposed model predicted the actual bond strength of a rebar embedded in PVA cementitious composites with or without CNTs well.

For the detailed investigation, the bond stress-slip responses measured through the test were compared with the proposed and previous models. As is compared in Figure 13, it was investigated that the previous models overestimated the bond stress overall, and they overestimated more with the increase in the rebar diameter. On the other hand, the proposed model represented the actual bond stress-slip response, in general, well, although there was some deviation between the predictions and the test results. Therefore, the proposed model helps to predict the bond behavior of a rebar embedded in PVA cementitious composites with CNTs.

## 7. Conclusions

In this study, a total of 36 pull-out specimens were tested to investigate the bond behavior of a rebar embedded in PVA cementitious composites with or without CNTs. The test program considered the rebar diameter and CNTs mix ratio as the test variables. From the test results, the effect of the test variables was investigated on the bond stress-slip behavior, including the bond strength and the corresponding slip. To reasonably represent the bond behavior of a rebar embedded in PVA cementitious composites, a new simple proposed model has been proposed in this study. The proposed model has been verified through the comparison with the test results. The main results obtained through this study can be summarized as follows:All of the specimens exhibited pull-out failure with neither a splitting crack nor rebar yielding, so the bond behavior of the rebar embedded in the PVA cementitious composites could be rigorously measured.Regarding the effect of the rebar, the bond strength of the rebar embedded in PVA cementitious composites increased as the rebar diameter increased. Meanwhile, it was demonstrated that the slip corresponding to the bond strength increased as the rebar diameter increased.The bond strength of the rebar embedded in PVA cementitious composites generally increased with the increasing CNTs mix ratio because the compressive strength of the PVA cementitious composites was increased. Therefore, it can be concluded that the bond strength of a rebar embedded in PVA cementitious composites can be improved with CNTs.The existing models overestimated the test results for the bond behavior of the rebar embedded in PVA cementitious composites with no CNTs. The main reason for the overestimation is inferred to be because the existing models are designated for the bond behavior of a rebar embedded in ordinary concrete containing coarse aggregate. In contrast, the PVA cementitious composites contain no coarse aggregate.By comparing the test results and the existing models, it was shown that the existing models generally overestimated the bond strength of a rebar embedded in PVA cementitious composites by 42~66% as they were initially designated for a rebar embedded in ordinary concrete with coarse aggregated. In addition, the previous models generally overestimated the bond stress of a rebar embedded in cementitious composites, and this tendency was more severe as the rebar diameter increased.Through the regression with the test results, a new simple model has been proposed to represent the bond stress-slip behavior of a rebar embedded in PVA cementitious composites with or without CNTs. The bond strength and the corresponding slip were evaluated in the proposed model considering the rebar diameter. The effect of the CNTs’ mix ratio was considered with the compressive strength of the PVA cementitious composites. Through the comparison with the test results, the proposed model predicted the actual bond strength of the rebar well, with an average of 1.02 and a standard deviation of 0.24 for the ratio of the model predictions to the test results.It is expected that the results of this study can be used in research on the anchorage length of a rebar and crack control in PVA cementitious composites with or without CNTs. In addition, this study can be helpful in the relevant research area on the structural behavior of PVA cementitious composites with or without CNTs.

## Figures and Tables

**Figure 1 polymers-15-00884-f001:**
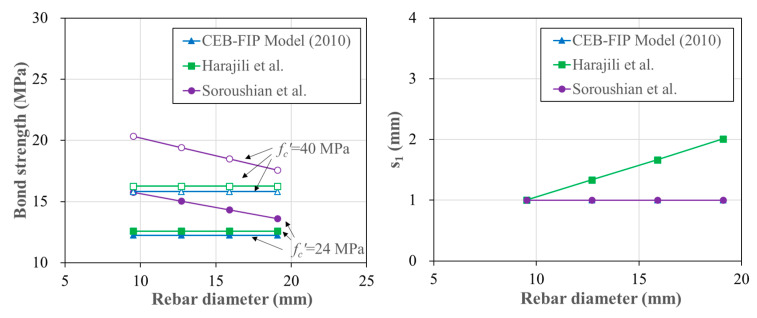
Comparison of the previous models on the bond strength and the slip corresponding to the bond strength [34,36,37].

**Figure 2 polymers-15-00884-f002:**
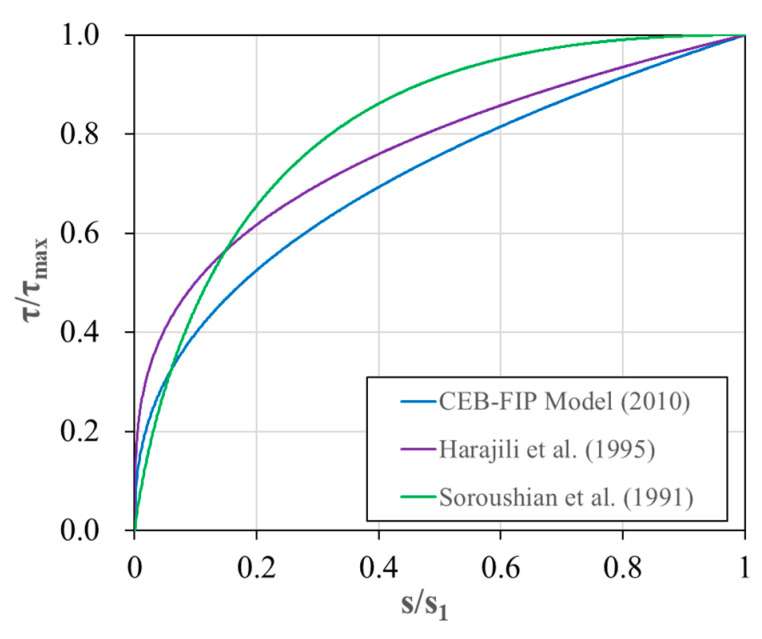
Normalized bond stress-slip curve for the ascending part of the previous models [34,36,37].

**Figure 3 polymers-15-00884-f003:**
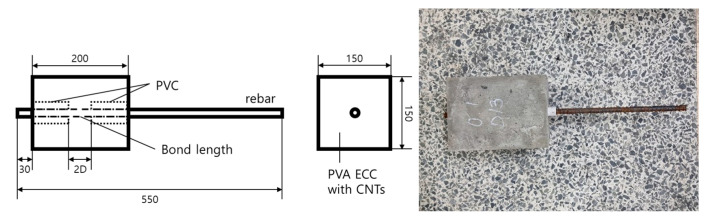
Pullout test specimen and its geometrical configuration.

**Figure 4 polymers-15-00884-f004:**
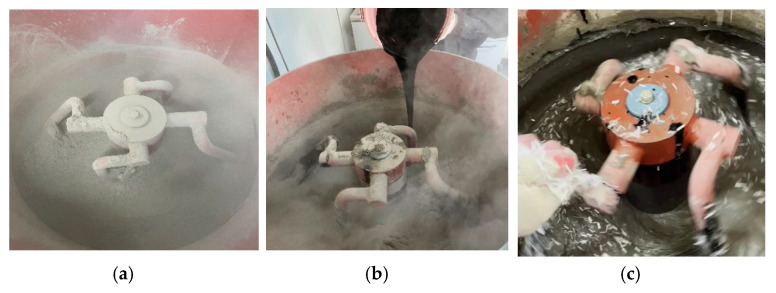
Mix procedure of PVA cementitious composites with CNTs. (**a**) Dry mix, (**b**) Addition of water and CNTs, (**c**) Addition of PVA fibers.

**Figure 5 polymers-15-00884-f005:**
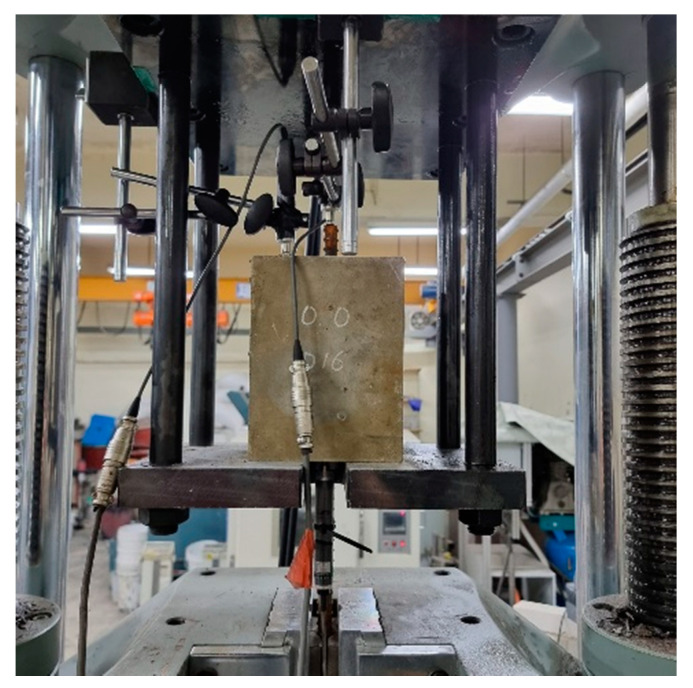
Pull-out test setup.

**Figure 6 polymers-15-00884-f006:**
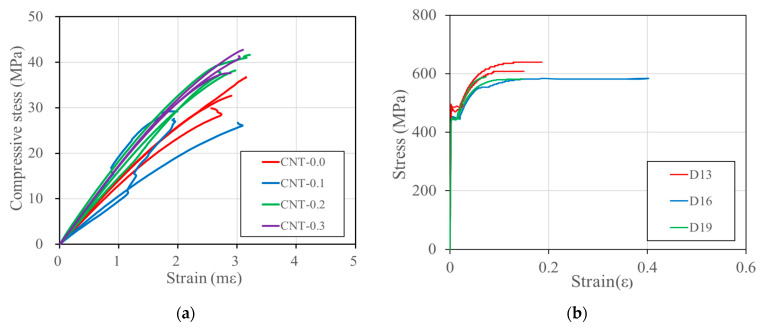
Stress-strain relations of the materials. (**a**) PVA cementitious composites with CNTs, (**b**) the rebars.

**Figure 7 polymers-15-00884-f007:**
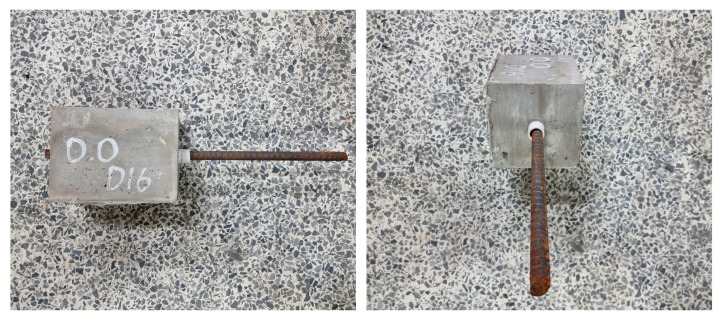
Pull-out test failure observed after the test.

**Figure 8 polymers-15-00884-f008:**
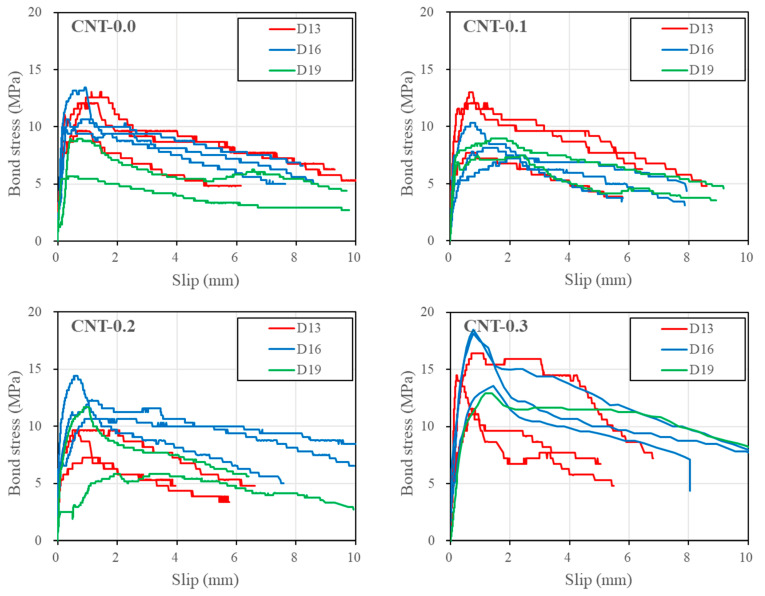
Bond stress-loaded end slip responses of a rebar embedded in PVA cementitious composites with CNTs.

**Figure 9 polymers-15-00884-f009:**
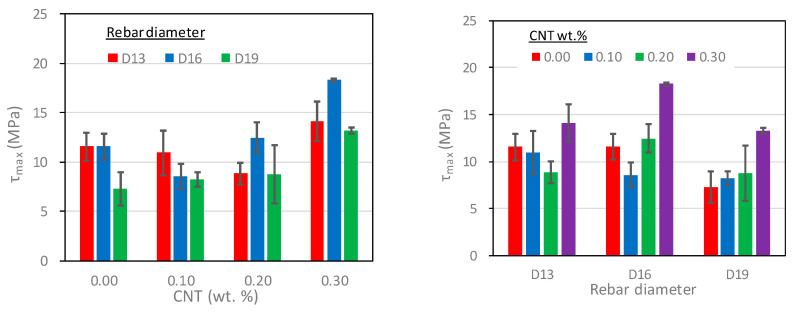
The effect of CNTs mix ratio and rebar diameter on the rebar bond strength.

**Figure 10 polymers-15-00884-f010:**
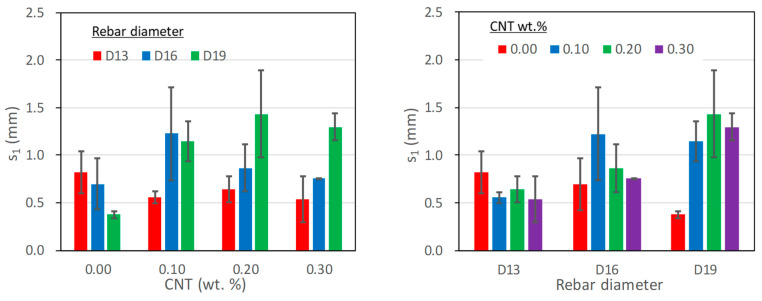
The effect of CNTs mix ratio and rebar diameter on the slip corresponding to the bond strength.

**Figure 11 polymers-15-00884-f011:**
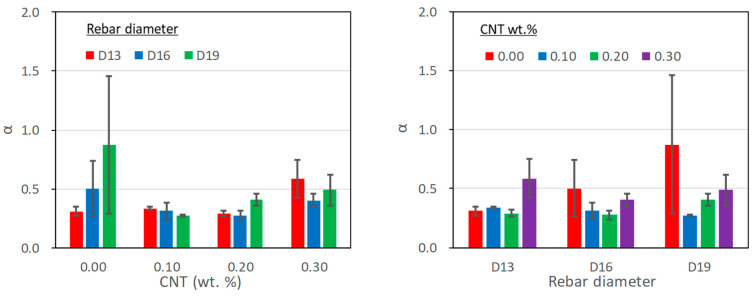
The effect of CNTs mix ratio and rebar diameter on α coefficient.

**Figure 12 polymers-15-00884-f012:**
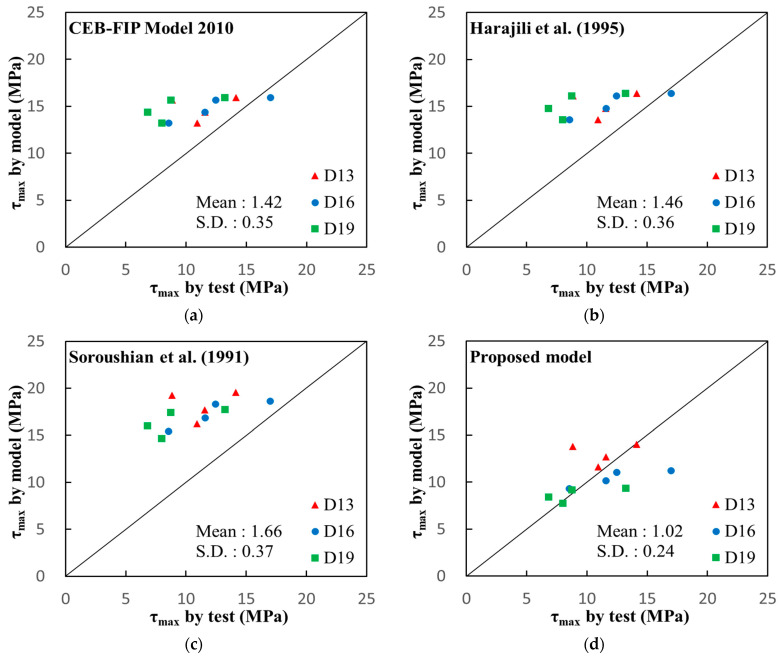
Comparison of the test results and the models on the bond strength of rebar embedded in PVA cementitious composites with CNTs: (**a**) CEB-FIP Code Model 2010 [37] (**b**) Harajili et al. (1995) [36] (**c**) Soroushian et al. (1991) [34] (**d**) Proposed model.

**Figure 13 polymers-15-00884-f013:**
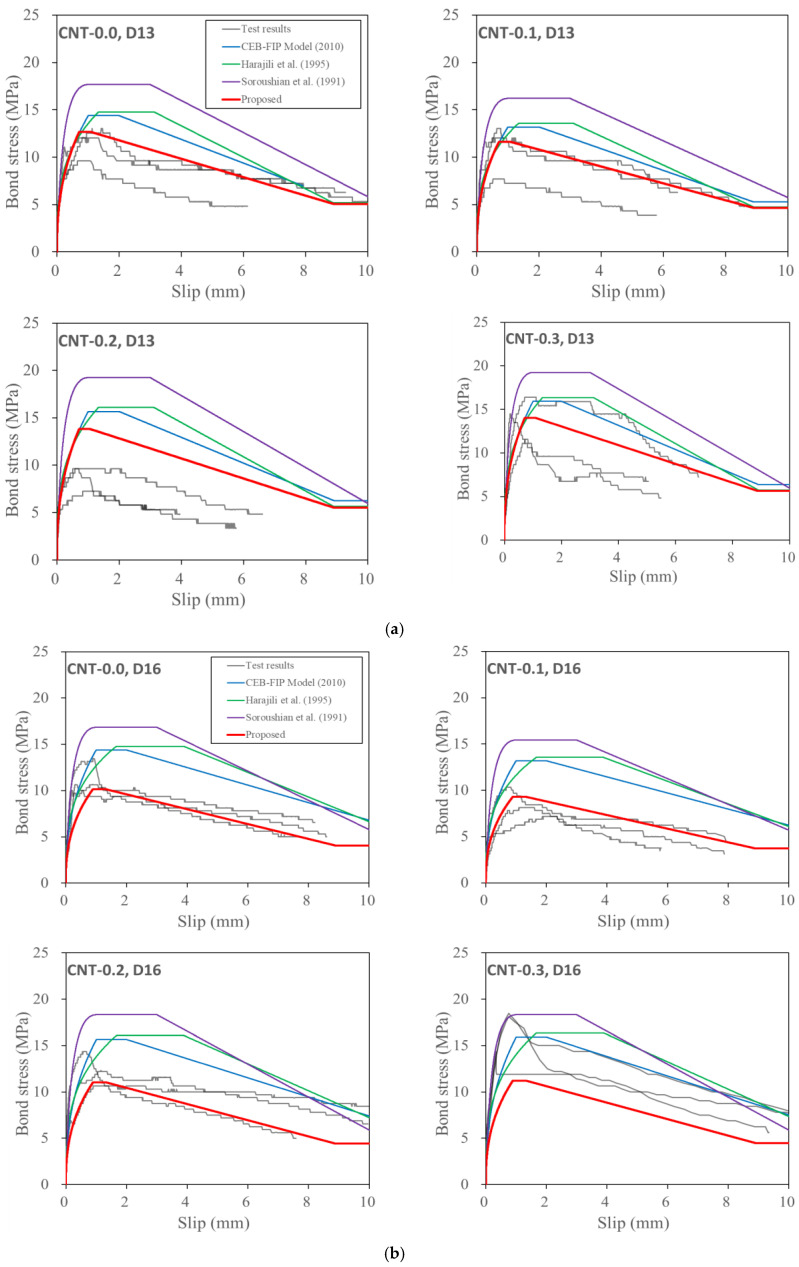
Comparison of the test results and bond stress-slip model: (**a**) D13 (**b**) D16 (**c**) D19, [34,36,37].

**Table 1 polymers-15-00884-t001:** Differences between SWCNTs and MWCNTs.

Type	Diameter (nm)	Length (nm–cm)	Tensile Strength (GPa)	Electrical Conductivity (S/cm)	Heat Conductivity (W/m·k)
SWCNTs	0.5–3.0	100–1	Up to 53	10,000	Max. 6000
MWCNTs	5–100	100–1	Up to 63	6000	Max. 3000

**Table 2 polymers-15-00884-t002:** Mix proportions of PVA cementitious composites with CNTs.

Specimen	CNT (wt.%)	Binder (B) (kg/m^3^)			Nonbinder (kg/m^3^)				Water (W) (kg/m^3^)	W/B (%)	Fiber Volume Fraction (%)
		OPC	BFS	FA	Sand	LA	SP	SRA			
CNT-0.0	0.0	412	220	412	275	14	1.92	0.4	343	32.9	2.07
CNT-0.1	0.1										
CNT-0.2	0.2										
CNT-0.3	0.3										

**Table 3 polymers-15-00884-t003:** Properties of PVA fibers.

Length (mm)	Diameter (mm)	Density (g/cm^3^)	Tensile Strength (MPa)	Modulus of Elasticity (MPa)
12	0.039	1.3	1600	25~40

**Table 4 polymers-15-00884-t004:** Properties of steel rebars.

Type	Nominal Diameter (mm)	Nominal Cross-Sectional Area (mm^2^)	Nominal Perimeter (mm)	Rib Distance (mm)	Rib Height (mm)		Yield Strength (MPa)	Elastic Modulus (GPa)
					Min.	Max.		
D13	12.7	126.7	40	8.9	0.5	1.0	484	18.8
D16	15.9	198.6	50	11.1	0.7	1.4	447	19.9
D19	19.1	286.5	60	13.4	1	2	444	20.9

**Table 5 polymers-15-00884-t005:** Slump and slump flow.

Specimen	Slump (mm)	Slump Flow (mm)
CNT-0.0	28.5	57.0
CNT-0.1	28.0	49.0
CNT-0.2	21.5	35.0
CNT-0.3	11.0	29.0

**Table 6 polymers-15-00884-t006:** Summary of the compression test results.

Specimen	Compressive Strength (MPa)		Strain (×10^−3^)		Modulus of Elasticity (GPa)	
	Each	Average (S.D.)	Each	Average (S.D.)	Each	Average (S.D.)
CNT-0.0	36.8	33.1 (2.8)	3.15	2.87 (0.24)	14.2	13.9 (0.5)
	32.7		2.90		14.2	
	29.9		2.56		13.2	
CNT-0.1	27.6	27.8 (2.3)	1.94	2.26 (0.54)	9.6	12.4 (3.2)
	29.2		1.82		16.9	
	26.7		3.01		10.6	
CNT-0.2	37.9	39.2 (2.9)	2.69	2.89 (0.14)	14.7	16.5 (1.5)
	38.2		2.97		16.4	
	41.7		3.01		18.4	
CNT-0.3	42.8	40.6 (3.0)	3.10	2.99 (0.12)	17.4	17.4 (0.2)
	41.4		3.03		17.3	
	37.6		2.82		17.7	

**Table 7 polymers-15-00884-t007:** Summary of the pullout test results.

Specimen	Rebar Type	Bond Strength $\tau_{max}\;(\mathrm{MPa})$		Slip Corresponding to the Bond Strength, $s_{1}\;(\mathrm{mm})$		α Coefficient	
		Each	Average (S.D.)	Each	Average (S.D.)	Each	Average (S.D.)
CNT-0.0	D13	12.1	11.8 (1.4)	0.67	0.82 (0.22)	0.32	0.31 (0.04)
		9.6		0.67		0.26	
		13.0		1.13		0.35	
	D16	10.7	11.6 (1.3)	0.32	0.70 (0.27)	0.84	0.50 (0.24)
		10.7		0.87		0.34	
		13.5		0.90		0.33	
	D19	5.6	7.3 (1.7)	0.34	0.38 (0.04)	0.29	0.88 (0.59)
		9.0		0.41		1.46	
CNT-0.1	D13	12.1	10.9 (2.3)	0.51	0.56 (0.06)	0.33	0.34 (0.01)
		7.7		0.52		0.35	
		13.0		0.64		0.33	
	D16	8.2	8.6 (1.3)	1.13	1.22 (0.49)	0.39	0.31 (0.07)
		10.3		0.68		0.33	
		7.2		1.86		0.22	
	D19	7.5	8.2 (0.7)	0.93	1.14 (0.21)	0.28	0.28 (0.01)
		9.0		1.36		0.27	
CNT-0.2	D13	9.6	8.8 (1.1)	0.51	0.64 (0.14)	0.31	0.29 (0.03)
		7.3		0.83		0.25	
		9.6		0.59		0.31	
	D16	10.7	12.5 (1.5)	0.90	0.87 (0.25)	0.33	0.28 (0.04)
		12.3		1.16		0.25	
		14.4		0.55		0.25	
	D19	11.7	8.8 (2.9)	0.98	1.43 (0.46)	0.36	0.41 (0.05)
		5.8		1.89		0.46	
CNT-0.3	D13	11.6	14.1 (2.0)	0.43	0.54 (0.24)	0.43	0.59 (0.16)
		16.4		0.52		0.52	
		14.5		0.81		0.81	
	D16	18.5	18.3 (0.2)	0.75	0.75 (0.00)	0.35	0.41 (0.05)
		18.1		0.76		0.46	
	D19	13.5	13.2 (0.3)	1.44	1.30 (0.14)	0.36	0.49 (0.13)
		12.9		1.16		0.62	

**Table 8 polymers-15-00884-t008:** Comparison on the bond strength between the test results and the models.

	Model		CEB-FIP Model Code 2010 [37]		Harajili et al. [36]		Soroushian et al. [34]		Proposed Model	
Rebar	Specimen	Test	Model	M/T	Model	M/T	Model	M/T	Model	M/T
**D13**	CNT-0.0	11.6	14.4	1.24	14.8	1.28	17.7	1.53	12.7	1.10
	CNT-0.1	10.9	13.2	1.21	13.6	1.24	16.2	1.48	11.6	1.06
	CNT-0.2	8.84	15.7	1.77	16.1	1.82	19.2	2.18	13.8	1.56
	CNT-0.3	14.1	15.9	1.13	16.4	1.16	19.6	1.38	14.0	0.99
**D16**	CNT-0.0	11.6	14.4	1.24	14.8	1.28	16.8	1.45	10.1	0.87
	CNT-0.1	8.55	13.2	1.54	13.6	1.59	15.4	1.80	9.29	1.09
	CNT-0.2	12.5	15.7	1.26	16.1	1.29	18.3	1.47	11.0	0.89
	CNT-0.3	17.0	15.9	0.94	16.4	0.96	18.6	1.10	11.2	0.66
**D19**	CNT-0.0	6.81	14.4	2.11	14.8	2.17	16.0	2.35	8.43	1.24
	CNT-0.1	7.99	13.2	1.65	13.6	1.70	14.7	1.84	7.73	0.97
	CNT-0.2	8.75	15.7	1.79	16.1	1.84	17.4	1.99	9.18	1.05
	CNT-0.3	13.2	15.9	1.20	16.4	1.24	17.7	1.34	9.34	0.71
**Avg.**				1.42		1.46		1.66		1.02
**S.D.**				0.35		0.35		0.37		0.24

## Data Availability

Data including the test results can be accessed through the corresponding author.

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
