# Peer review of "Bond Behavior of Steel Rebar Embedded in Cementitious Composites Containing Polyvinyl Alcohol (PVA) Fibers and Carbon Nanotubes (CNTs)"

_polymers, 2023, doi:10.3390/polym15040884_

Round 1

Reviewer 1 Report

The study attempts a detailed experimental and analytical investigation on the behavior of PVA fiber reinforced cementitious matrix. However, the limitations of the present work is of two folds:

1. The paper is limited only understanding the bond behavior of PVA based cementitious system which is not sufficient for a standard journal article. Moreover, the test parameters considered in the study are very limited and not exhaustive.

2. What is the novelty of the proposed work? A number of previous studies have already attempted the use of PVA based system. If the authors claim that they study the interaction between PVA and CNC system, then they fail to do it by a large margin.

3. What is the need for a new analytical model since number of previous bond stress models are suggested by the code itself. It looks like a new model is proposed for the sake of proposing. If the model is really essential, a proper scientific justification is to be provided.

4. In continuation with the above comment, the validity of the proposed model is to be verified with the existing literature works. Also, a detailed parametric study is essential.

5. Failure mode images presented in Figure 6 does not apparently highlight the type of failure observed. Provide better images for the same. Moreover, improve the conclusion section which looks like a mere summary.

For the above reasons, the paper cannot be considered for publication in present form and requires significant enhancement or revisions. Hence, the manuscript is to be declined with no encouragement to re-submit.

Author Response

The authors appreciate the comments of the reviewer.

Please see the authors’ responses to the reviewer’s comments in the attached file.

Reviewer 2 Report

The article comprehensively discusses the bond behavior of steel rebars with cementitious composites to which CNT is added in different rates. 

In general, the scope of the article may be limited, but it is well developed, both in its methodology and in its results and conclusions. In addition, the quality of the descriptions and graphics is high. The conclusions correctly summarize the work carried out. 

From a scientific point of view, I wonder if the design approach of the research, in the sense of maintaining the same mix design for all the mixtures, is adequate, since cementitious composites are being developed with very different mechanical qualities (compressive strength and slump, see table 5), so I wonder to what extent they are comparable. This would be for me the main weakness of the article. 

In my opinion, the first 6 lines of the ABSTRACT are superfluous because they are generic. I would delete them or reduce them to a minimal. On the other hand, I consider it relevant to add in the abstract that the cementitious composite includes OPC, BFS and FA. 

The INTRODUCTION, although brief, is well developed. However, certain changes need to be made in this regard: 

  • - Firstly, the references should be included together with their main findings, for a better understanding of the general state of the art. For example (and not exhaustively), references 22, 24 and 25 are only cited, their main outcomes should be included. Please, check.  

  • - Secondly, there is an excess of self-citations in the text. Please cite your previous work only when it is directly related to the present work. Citations of generic statements (such as references 6,7,8) must be eliminated. 

  • - Third, please eliminate brackets with multiple citations within them without elaborating on their main findings, such as [2-8] or [9-13]. 

  • - Finally, my recommendation would be to make a subsection of the introduction with specifically the background of this research, whether or not it is already published. It is important to contextualize this article (pull-out tests) within the research that you must have done on PVA cementitious composites with CNTs (the rest of the investigation project).  

  •  

As for the METHODOLOGY (materials and tests), it is written in a very comprehensive way. 

Regarding the RESULTS AND CONCLUSIONS, some explanation should be given as to why existing models overestimate the results even in the case of CNT=0%. Is it due to the fibers? Could it be due to the unconventional binders used? 

Some additional MINOR REMARKS:  

  • - Section 2.2 is missing 

  • - Figure 4 and 5 could be presented together to maximize the space, or even deleted, as they do not share important information.  

  • - In figure 11 part of the label is missing. 

  • - Figure 14 needs to be improved in size or quality.

Author Response

The authors appreciate the expert comments of the reviewer.

Please see the authors’ responses to the reviewer’s comments in the attached file.

Reviewer 3 Report

Bond behaviour of steel rebar embedded in PVA cementitious composites with CNTs

The title contains acronyms that should be written in full.

My suggestion however is also that the title is amended to:

Bond behaviour of steel rebar embedded in cementitious composites also containing polyvinyl alcohol fibres (PVA) and carbon nanotubes (CNT)

American spelling has been used throughout the paper.

The use of English could be improved in many areas.

Most of the Abstract should be redrafted and meaningless text such as, line 2: 'Meanwhile, lots of....' should be deleted.

The overall layout of the paper is very poor and requires major revision.

The Introduction is on the long side and much of it should be in a section under 'Literature Review'.

A section on the 'Research Significance' of this study should be included.

I would suggest therefore that the paper is amended and redrafted and include:

1.0 Introduction

2.0 Literature Review (This should contain some of the Introduction and the whole of the existing section 4.0.)

3.0 Research Significance (This section should also include the objectives of the test program.)

4.0 Test Program (Currently existing section 2.0.)

5.0 Test Results (Currently existing section 3.0.)

6.0 Discussion of Results (Currently existing section 5.0.)

7. 0 Conclusions (Currently existing section 6.0.)

Other comments on existing text:

Table 2: The concrete mix contents are the same except for the variation in CNT wt %. Therefore, the Table should not show the same repeated values. These need to be only shown once. The Table therefore needs to be amended.

All acronyms should be written in full in all of the Tables.

Existing section 2.0 should be reorder:

2.1 Materials (numbering to remain the same)

2.2  - missing?

2.3 Pullout test specimens (numbering to remain the same)

2.4 Fabrication (should be renumbered as section 2.2)

2.5 Pull-out test set-up and procedures (should be renumbered as section 2.4)

Table 5 contains slump and slump flow - these values should be in existing section 2.4. All acronyms should be written in full in the Table. All acronyms should be written in full in the Table.

Figure 4 is far too small and should be either increased in size or colour combinations changed.

Ditto my comments for Figures 5 and 7.

Figure 8 (a), Figure 9 (a) and Figure 10 (a) - not clear what these are provided for as Figure 8 (b), Figure 9 (b) and Figure 10 (b) contain the same information. Text 'CNT wt %' needs to be added to these parts (b). This information is also provided in Table 6.

Conclusions 2) and 3), lines 1, mostly contain the same repeated text, Therefore, 3) should be amended.

Unfortunately, I am unable to recommend publication of this paper with major revisions being carried out.

Author Response

(The authors gave the same response as above.)

Round 2

Reviewer 1 Report

Most of the comments were addressed in the revised submission.

Author Response

The authors appreciate the comments of the reviewer.

Reviewer 3 Report

A number of major improvements have been made to this article.

My suggestions are as follows:

Abstract

Delete new lines 23 and partial line 24, starting sentence with ‘This study…..’ Include CNTs in full in line 25.

The Introduction is still far too long.

New lines 48 to 119 should be included in the new section 2, Literature Review as these describe what other researchers have done and found etc. This is not appropriate in the Introduction.

Table 1 appears to be the same as the previous version. Not clear what has changed.

Following these changes to the changes already made, I would recommend publication.

Author Response

The authors appreciate the expert comments of the reviewer. Please see the attached file containing the authors’ responses to the reviewer’s comments.
